# Dess–Martin Periodinane-Mediated Oxidative Coupling Reaction of Isoquinoline with Benzyl Bromide

**DOI:** 10.3390/molecules28030923

**Published:** 2023-01-17

**Authors:** Chunmei Yang, Guoqing Zhang, Senling Tang, Yang Pan, Huawu Shao, Wei Jiao

**Affiliations:** 1Natural Products Research Centre, Chengdu Institute of Biology, Chinese Academy of Sciences, Chengdu 610041, China; 2University of Chinese Academy of Sciences, Beijing 100049, China

**Keywords:** hypervalent iodine, Dess–Martin periodinane, isoquinoline, isoquinolinone, benzyl bromide

## Abstract

Dess–Martin periodinane (DMP) is a broadly applicable oxidant in chemical synthesis. In this work, an efficient and convenient synthesis of *N*-substituted isoquinolinone derivatives mediated by DMP was achieved through the oxidative coupling reaction of functionalized isoquinoline with readily available benzyl bromide, which is a metal-free, mild, and practical method for synthesizing isoquinoline-1,3-dione or isoquinoline-1,3,4-trione derivatives in excellent yields. The H_2_O^18^-labeling experiment was performed to gain insight into the possible mechanism for this reaction.

## 1. Introduction

Isoquinoline is an essential structural unit in pharmaceutical molecules and naturally occurring alkaloids [1,2,3]. In particular, isoquinolinones and their derivatives are the critical moiety of many pharmaceuticals [4,5]. As a member of the isoquinoline alkaloid family, isoquinolinones have exhibited a wide range of structural diversity and biological activity [6,7], and gliquidone is used as a commercial drug to treat diabetes (Figure 1, **I**) [8]. It was also found that isoquinolinones exhibit antiviral (Figure 1, **II**) [9], anticancer (Figure 1, **III**) [10], and anti-metastatic biological activities (Figure 1, **IV**) [11].

Various methods have been developed to synthesize isoquinolinone derivatives due to their significant pharmacological properties and diverse biological activities. Specifically, several synthetic strategies, such as organic electrochemistry [12], visible-light catalysis [13], and metal-catalyzed strategy [14], have been used to initiate these intriguing reactions and have proven highly useful. It was reported that a mild and efficient carboperfluoroalkylation reaction of alkenes led to isoquinoline-1,3-diones using perfluoroalkyl iodides or bromides (Figure 2a) [15]. The synthesis of nitro-functionalized isoquinolinedione derivatives has also been achieved by a tandem reaction (Figure 2b) [16]. In 2012, Da-you Ma et al. developed a convenient iodine-catalyzed multiple C-H bonds functionalization of isoquinolines with methylarenes to produce *N*-benzyl isoquinoline-1,3,4-triones (Figure 2c) [17]. There are still some limitations, although these methods have synthesized the target product, such as high temperature and heavy metal residue. In particular, the direct synthesis of isoquinoline-1,3-dione from isoquinoline as the substrate has been rarely reported [18]. Therefore, it is necessary and valuable to develop a more practical and efficient method to synthesize isoquinoline-1,3-dione.

The oxidative coupling reaction mediated by hypervalent iodine reagents (HIRs) have the advantages of mild reaction conditions, high selectivity, and environmental friendliness compared with traditional metal catalysts [19]. HIRs have become a powerful tool for new bond formation due to their excellent ability to construct C-C and carbon–heteroatom bonds [20]. In recent decades, hypervalent iodine (V) reagents have become a powerful synthetic tool in organic chemistry [21,22]. In particular, DMP has been proven an effective oxidant to construct C-N bonds [23].

In the previous study, our group disclosed a method of direct trifluoromethylation and perfluoroalkylation of isoquinoline mediated by [Bis(trifluoroacetoxy)iodo]benzene (PIFA) [24], which led us to speculate that hypervalent iodine may induce isoquinoline to undergo C-H functionalization. Herein, as part of our continuing research on the functionalization of isoquinoline caused by hypervalent iodine reagent, we first disclose a metal-free oxidative transformation of isoquinoline to provide isoquinolinone with DMP as the oxidant. Compared with the traditional synthesizing isoquinolinone, this method has the edge of easy substrate acquisition, mild reaction conditions, one step, and environmental friendliness (Figure 2d).

## 2. Results

Based on the above speculation, the model substrate of isoquinoline (**1a**) with benzyl bromide (**2a**) was first examined. Then, they were treated with 2.0 eq. of DMP in *N*-methyl-2-pyrrolidone (NMP) at room temperature. After 24 h, it was found that the desired oxidative coupling product **3aa** could be obtained in 62% yield, along with a trace amount of oxylated product **4** forming (Table 1, entry 1). The structure of **3aa** was confirmed by X-ray single-crystal diffraction (Figure 3). When DMP was replaced by other hypervalent iodine reagents, such as (diacetoxyiodo)benzene (PIDA), PIFA, Koser’s reagent, and iodosobenzene (PhIO), no reaction occurred, and the starting material **1a** was almost recovered (Table 1, entries 2−5). When 2-iodoxybenzoic acid (IBX) was used, the desired product **3aa** was obtained in 47% yield, a little lower than the yield of DMP (Table 1, entry **6**). Different solvents were tested with hypervalent iodine (V) reagent DMP to enhance the yield of product **3aa**. NMP was found to serve as the best reaction medium as compared to the other solvents, such as dimethylformamide (DMF), acetonitrile (MeCN), and dimethyl sulfoxide (DMSO) (Table 1, entry 7–9). In addition, the hypervalent iodine (III) reagent could successfully catalyze the reaction when the solvent was DMF, and the undesired isoquinoline-1-one **4** was formed along without the product **3aa** (Table 1, entry 10).

We further studied the effects of reaction temperature, substrate equivalence ratio, and water content of the reaction system on the reaction yield. When the temperature decreased, the reaction took more time, and the yield also decreased (Table 1, entry 12). Higher temperatures also led to lower yield (Table 1, entry 13). If the temperature dropped to −10 °C, isoquinoline did not react with benzyl bromide (Table 1, entry 11). Further investigation of the amount of DMP revealed that 3.4 eq. of DMP was enough for the present reaction (Table 1, entry 15). In addition, the controlled experiments proved that the reactions must be conducted under air. Firstly, no corresponding products **3aa** or **4** were detected with substrate **1a** and 3.4 eq. of DMP in NMP at room temperature under N_2_ for 36 h (Table 1, entry 16). Conversely, the increase in the ten equivalents of H_2_O led to a decrease in yield (Table 1, entry 17). Hence, trace amounts of H_2_O in the air are sufficient for the process without additional water. Due to the results presented above, the optimal conditions for product **3aa** were DMP (3.4 eq.) in NMP at room temperature under air for 36 h.

With the optimized reaction conditions in hand (Table 1, entry 15), the substrate scope was investigated. We investigated the generality of this multiple C-H bonds functionalization of isoquinoline with benzyl bromide. The substrate scope for the benzyl bromide moiety is listed in Figure 4. Benzyl bromides bearing electron-donating substituents were successfully tolerated to afford the corresponding *N*-substituted isoquinoline-1,3-diones in excellent yields, such as methyl (**3ab**, **3ac**) and methoxy (**3ad**, **3ae**) groups. As expected, 3-chloro and 3-bromo -substituted benzyl bromides were suitable substrates for this transformation, which all reacted with isoquinoline smoothly to yield corresponding isoquinoline-1,3-diones (**3af**, **3ag**). Meanwhile, when one strong electron-withdrawing nitro group was installed in the 4-position of benzyl bromide, the desired product **3ah** could be obtained in 78% yield.

Besides benzyl bromide derivatives, the optimized reaction condition was further applied to allyl bromide and 2-(bromomethyl) naphthalene. For an allyl bromide, the oxidative coupling reaction gave the desired product **3ai** in 65% yield. Secondly, the oxidative coupling reaction of 2-(bromomethyl) naphthalene smoothly reacted with isoquinoline to provide the corresponding product **3aj** as a white solid in 73% yield. The corresponding isoquinolinone was not detected when 1-bromopentane was stirred with isoquinoline **1a**.

Encouraged by the exciting results obtained from the hypervalent iodine-catalyzed multiple C-H bonds functionalization of isoquinoline with various benzyl bromides, we then focused on the scope of isoquinoline derivatives (Figure 5). As expected, 6-methyl, 6-chloro, and 6-bromo -substituted isoquinolines were suitable substrates for this transformation, which all reacted with benzyl bromide smoothly to yield substituted *N*-benzyl isoquinoline-1,3-diones (**3ba**–**3bh**). Furthermore, when bromine was installed in the 7-position of isoquinoline, benzyl bromide was consumed, providing the desired isoquinoline-1,3-diones (**3bi**, **3bj**). However, substrates were not converted when one strong electron-withdrawing nitro group was installed in isoquinoline, and the related product was not observed.

In further studies, the results revealed that the substituents in different positions and types greatly influenced the structure of the products (Figure 6). A similar phenomenon has also been reported by Xu’s group [25]. Firstly, the optimized reaction conditions were further applied to the reaction of 5-bromoisoquinoline with benzyl bromides, but the corresponding isoquinoline-1,3-diones cannot be detected, with *N*-benzylation product (isoquinoline-1,3,4-triones, **3ca**) at around 59% yield. Similarly, 5-chloroisoquinoline exhibited a similar property, providing isoquinolinetrione at 61% yield (**3cb**). Secondly, when one strong electron-donating methoxy group was installed in the 5-, 6-, or 7-position of isoquinoline, no desired isoquinoline-1,3-diones were formed, whereas isoquinoline-1,3,4-triones were afforded in medium-to-excellent yields (**3cc**–**3cg**).

Many studies have proved that DMP is less likely to provide oxygen atoms in oxidation reactions [26,27,28], as the oxygen atoms in the product may come from water in the air. An H_2_O^18^-labeling experiment was carried out with 3.4 eq. of DMP in a mixed solvent consisting of NMP and H_2_O^18^ at room temperature under N_2_ for 36 h (Figure 7). The corresponding molecular weight (**3da**) was detected by ESI. The result indicates that the oxygen at the C-1 or C-3 position of the corresponding isoquinolinones stemmed from H_2_O.

## 3. Discussion

Based on the mechanistic experiments described above and previous studies [25,29,30,31,32], a plausible mechanism for this hypervalent iodine-catalyzed multiple C-H bonds functionalization is proposed in Figure 8, with the reaction of isoquinoline (**1a**) and benzyl bromide (**2a**) shown as an example. Initially, the quaternary ammonium salt **A** is generated from benzyl bromide reacted with isoquinoline readily. Subsequently, **A** is attacked by H_2_O, followed by the removal of one hydrobromide (HBr), affording hydroxyl intermediate **B**. DMP easily oxidizes the intermediate **B** to obtain enamide **C**, and DMP is reduced to iodinane **INT-1**. Referring to Robert H. Dodd’s research [33], when the bromide anions are present, the iodinane **INT-1** reacts with the bromide source in situ to generate **INT-2**. The intermediate **D** is obtained via the bromophilic attack of **C** on hypervalent iodine **INT-2**. The key intermediate 2-benzyl-4-bromoisoquinolinone **E** is then formed via isomerization. Intermediate **E** reacts with another equivalent of hypervalent iodine **INT-2** to form iodonium salt **F**. Bromine anion attacks the C-2 position of iodonium salt **F**, followed by the removal of one iodinane, to obtain **G**, whereafter the nucleophilic attack of H_2_O in the C-3 of isoquinolinone generates hydroxyl product **H**, which is easily oxidized by the third equivalent of DMP to obtain **3aa**.

On the other hand, when methoxyl is presented or halogen-substituted at C-5 of isoquinoline, for H_2_O it may be easier to nucleophilic attack intermediate iodonium salt **F** than bromine. Therefore, **F** was attacked by H_2_O at the C-3 position to obtain **I**. Then, the water acts as a nucleophile and attacks the second iodonium salt **I** at the C-4 position to further convert to **L** by losing one hydrogen bromide. Finally, intermediate **L** is oxidized by the third equivalent of **DMP** to generate the final isoquinoline-1,3,4-trione compound **3ca**.

## 4. Materials and Methods

### 4.1. Materials

All commercially sourced reagents were used as supplied unless otherwise stated. The ^1^H NMR and ^13^C NMR spectra were recorded on a Bruker Avance 400 spectrometer (400 MHz or 100 MHz, respectively). ^1^H NMR chemical shifts were reported in ppm (δ) relative to tetramethylsilane (TMS) with the solvent resonance employed as the internal standard (CDCl_3_, *δ* 7.26 ppm). ^13^C NMR chemical shifts were determined relative to the solvent: CDCl_3_ at *δ* 77.16 ppm. High-resolution mass spectroscopy data of the products were collected on a BioTOF Q instrument. Infrared (IR) spectra are obtained by the use of Spectrum One and expressed in wave number (cm^−1^). Visualization on TLC was achieved by UV light (254 nm). Flash chromatography was performed on silica gel 90, 200–300 mesh.

### 4.2. General Procedure for the Oxidative Coupling Reactions

To a solution of isoquinoline **1** (0.1 mmol) and benzyl bromide **2** (0.25 mmol) in NMP (6 mL), in a 10 mL rounded bottom flask was added DMP (0.34 mmol). The reaction was stirred at room temperature and monitored by TLC. The mixture was then allowed to stir at room temperature for 36 h. After the complete consumption of substrates (monitored by TLC), the resulting mixture was diluted with EtOAc and washed with saturated NaCl solution (×3). The organic layer was dried over Na_2_SO_4_ and concentrated under a vacuum. The crude product was purified by silica gel column chromatography using 5% EtOAc/petroleum ether as the eluent.

Please see the Appendix A for all synthesized compounds’ original ^1^H and ^13^C NMR spectra. Detailed physicochemical properties of isoquinolinone derivatives:**2-Benzyl-4,4-dibromoisoquinoline-1,3(2H,4H)-dione (3aa)**. White solid, 193 mg; mp: 116 °C; IR (KBr) cm^−1^ 2965, 1679, 1597, 1429, 1375, 1336, 1271, 1238, 1150, 960,731, 699; ^1^H NMR (400 MHz, CDCl_3_) δ 8.14 (t, *J* = 7.6 Hz, 2H), 7.75 (t, *J* = 7.8 Hz, 1H), 7.54 (t, *J* = 7.7 Hz, 1H), 7.48 (d, *J* = 7.4 Hz, 2H), 7.30 (dt, *J* = 19.2, 7.2 Hz, 3H), 5.26 (s, 2H). ^13^C NMR (100 MHz, CDCl_3_) *δ* 165.0, 161.5, 140.3, 135.4, 134.6, 130.4, 130.2, 128.7, 128.3, 127.6, 120.1, 44.9; ESI-HRMS: *m/z* calcd for C_16_H_11_Br_2_NO_2_Na [M + Na]^+^: 429.90542, found 429.90537.**4,4-Dibromo-2-(3-methylbenzyl)isoquinoline-1,3(2H,4H)-dione (3ab).** White solid, 190 mg; mp: 95 °C; IR (KBr) cm^−1^ 2960, 1724, 1681, 1598, 1428, 1371, 1347, 1235, 1150, 960, 748, 696; ^1^H NMR (400 MHz, CDCl_3_) *δ* 8.14 (ddd, *J* = 8.1, 7.1, 1.3 Hz, 2H), 7.74 (td, *J* = 7.7, 1.4 Hz, 1H), 7.53 (td, *J* = 7.6, 1.2 Hz, 1H), 7.28 (dd, *J* = 8.9, 1.9 Hz, 2H), 7.21 (t, *J* = 7.5 Hz, 1H), 7.09 (d, *J* = 7.5 Hz, 1H), 5.23 (s, 2H), 2.33 (s, 3H); ^13^C NMR (100 MHz, CDCl_3_) *δ* 165.3, 161.9, 140.6, 138.3, 135.7, 134.9, 130.7, 130.5, 129.6, 128.7, 128.6, 128.5, 125.9, 120.5, 51.0, 45.2, 21.5. ESI-HRMS: *m/z* calcd for C_17_H_13_Br_2_NO_2_Na [M + Na]^+^: 443.92107, found 443.92127.**4,4-Dibromo-2-(4-methylbenzyl)isoquinoline-1,3(2H,4H)-dione (3ac).** White solid, 201 mg; mp: 92 °C; IR (KBr) cm^−1^ 2960, 1726, 1690, 1598, 1438, 1372, 1337, 1307, 1237, 1154, 964, 746, 692; ^1^H NMR (400 MHz, CDCl_3_) *δ* 8.18–8.09 (m, 2H), 7.74 (td, *J* = 7.7, 1.5 Hz, 1H), 7.53 (td, *J* = 7.6, 1.3 Hz, 1H), 7.39 (d, *J* = 7.8 Hz, 2H), 7.13 (d, *J* = 7.8 Hz, 2H), 5.22 (s, 2H), 2.31 (s, 3H). ^13^C NMR (100 MHz, CDCl_3_) *δ* 165.0, 161.5, 140.3, 137.3, 134.5, 132.5, 130.4, 130.1, 128.9, 128.7, 128.3, 120.2, 44.6, 20.9; ESI-HRMS: *m/z* calcd for C_17_H_13_Br_2_NO_2_Na [M + Na]^+^: 443.92107, found 443.92129.**4,4-Dibromo-2-(3-methoxybenzyl)isoquinoline-1,3(2H,4H)-dione (3ad).** White solid, 197 mg; mp: 75 °C; IR (KBr) cm^−1^ 2962, 1724, 1683, 1609, 1508, 1457, 1376, 1333, 1246, 1183, 1026, 951, 748, 689; ^1^H NMR (400 MHz, CDCl_3_) *δ* 8.14 (td, *J* = 8.4, 1.3 Hz, 2H), 7.75 (td, *J* = 7.7, 1.5 Hz, 1H), 7.54 (td, *J* = 7.7, 1.2 Hz, 1H), 7.28–7.19 (m, 1H), 7.05 (d, *J* = 7.6 Hz, 1H), 7.01 (t, *J* = 2.1 Hz, 1H), 6.82 (dd, *J* = 8.1, 2.6 Hz, 1H), 5.23 (s, 2H), 3.78 (s, 3H). ^13^C NMR (100 MHz, CDCl_3_) *δ* 165.0, 161.5, 159.4, 140.3, 136.9, 134.6, 130.5, 130.2, 129.3, 128.3, 120.8, 120.0, 113.8, 113.4, 54.9, 44.7; ESI-HRMS: *m/z* calcd for C_17_H_13_Br_2_NO_2_Na [M + Na]^+^: 459.91599, found 459.91580.**4,4-Dibromo-2-(4-methoxybenzyl)isoquinoline-1,3(2H,4H)-dione (3ae).** White solid, 211 mg; mp: 63 °C; IR (KBr) cm^−1^ 2971, 1725, 1686, 1599, 1465, 1375, 1330, 1265, 1235, 1167, 1036, 955, 742, 712; ^1^H NMR (400 MHz, CDCl_3_) *δ* 8.13 (ddd, *J* = 7.9, 4.6, 1.2 Hz, 2H), 7.74 (td, *J* = 7.7, 1.4 Hz, 1H), 7.53 (td, *J* = 7.7, 1.2 Hz, 1H), 7.49–7.41 (m, 2H), 6.93–6.80 (m, 2H), 5.19 (s, 2H), 3.77 (s, 3H). ^13^C NMR (100 MHz, CDCl_3_) *δ* 165.0, 161.5, 159.0, 140.2, 134.5, 130.4, 130.2, 130.1, 129.1, 128.2, 127.6, 120.2, 113.6, 113.5, 54.9, 44.4; ESI-HRMS: *m/z* calcd for C_17_H_13_Br_2_NO_2_Na [M + Na]^+^: 459.91599, found 459.91613.**4,4-Dibromo-2-(3-chlorobenzyl)isoquinoline-1,3(2H,4H)-dione (3af).** Yellow oil, 190 mg; mp: 85 °C; IR (KBr) cm^−1^ 2920, 1720, 1670, 1595, 1419, 1369, 1338, 1266, 1236, 1158, 966, 748, 708; ^1^H NMR (400 MHz, CDCl_3_) *δ* 8.15 (td, *J* = 8.4, 1.3 Hz, 2H), 7.76 (td, *J* = 7.7, 1.4 Hz, 1H), 7.55 (td, *J* = 7.6, 1.2 Hz, 1H), 7.47 (q, *J* = 1.4 Hz, 1H), 7.37 (td, *J* = 4.7, 1.7 Hz, 1H), 7.32–7.16 (m, 2H), 5.22 (s, 2H). ^13^C NMR (100 MHz, CDCl_3_) *δ* 165.0, 161.5, 140.3, 137.9, 135.3, 134.6, 130.4, 130.1, 129.3, 128.4, 128.3, 128.2, 125.6, 120.2, 44.9; ESI-HRMS: *m/z* calcd for C_16_H_10_Br_2_ClNO_2_Na [M + Na]^+^: 463.86645, found 463.86648.**4,4-Dibromo-2-(3-bromobenzyl)isoquinoline-1,3(2H,4H)-dione (3ag).** Yellow oil, 187 mg; mp: 75 °C; IR (KBr) cm^−1^ 3074, 1727, 1671, 1595, 1436, 1376, 1336, 1272, 1234, 1153, 960, 884, 745, 708; ^1^H NMR (400 MHz, CDCl_3_) *δ* 8.13 (ddd, *J* = 10.3, 8.0, 1.3 Hz, 2H), 7.75 (td, *J* = 7.7, 1.4 Hz, 1H), 7.62 (t, *J* = 1.9 Hz, 1H), 7.54 (td, *J* = 7.6, 1.1 Hz, 1H), 7.44–7.36 (m, 2H), 7.18 (t, *J* = 7.9 Hz, 1H), 5.20 (s, 2H).^13^C NMR (100 MHz, CDCl_3_) *δ* 164.9, 161.4, 140.1, 137.5, 134.7, 131.5, 130.8, 130.5, 130.1, 129.8, 128.3, 127.2, 122.2, 119.9, 44.1. ESI-HRMS: *m/z* calcd for C_16_H_10_Br_3_NO_2_Na [M + Na]^+^: 507.81694, found 507.81615.**4,4-Dibromo-2-(4-nitrobenzyl)isoquinoline-1,3(2H,4H)-dione (3ah).** White solid, 211 mg; mp: 133 °C; IR (KBr) cm^−1^ 3061, 1725, 1686, 1600, 1517, 1430, 1380, 1345, 1235, 954, 705; ^1^H NMR (400 MHz, CDCl_3_) *δ* 8.20 (s, 1H), 8.21–8.11 (m, 3H), 7.79 (td, *J* = 7.8, 1.4 Hz, 1H), 7.68–7.61 (m, 2H), 7.57 (td, *J* = 7.6, 1.1 Hz, 1H), 5.33 (s, 2H). ^13^C NMR (100 MHz, CDCl_3_) *δ* 165.1, 161.6, 147.4, 142.4, 140.2, 135.0, 130.69, 130.3, 129.4, 128.4, 123.6, 119.7, 44.1; ESI-HRMS: *m/z* calcd for C_16_H_11_Br_2_N_2_O_4_ [M + H]^+^: 452.90856, found 452.90488.**2-Allyl-4,4-dibromoisoquinoline-1,3(2H,4H)-dione (3ai).** Colorless oil, 110 mg; mp: 63 °C; IR (KBr) cm^−1^ 2938, 1717, 1683, 1597, 1459, 1420, 1373, 1345, 1239, 1163, 965, 753, 699; ^1^H NMR (400 MHz, CDCl_3_) *δ* 8.14 (ddd, *J* = 10.8, 8.0, 1.3 Hz, 2H), 7.75 (td, *J* = 7.7, 1.4 Hz, 1H), 7.54 (td, *J* = 7.6, 1.2 Hz, 1H), 5.91 (ddt, *J* = 17.3, 10.2, 5.8 Hz, 1H), 5.34 (dq, *J* = 17.1, 1.5 Hz, 1H), 5.24 (dq, *J* = 10.3, 1.3 Hz, 1H), 4.66 (dt, *J* = 5.9, 1.4 Hz, 2H). ^13^C NMR (100 MHz, CDCl_3_) *δ* 164.6, 161.2, 140.3, 134.5, 130.4, 130.3, 130.2, 128.2, 120.0, 118.5, 43.6; ESI-HRMS: *m/z* calcd for C_12_H_10_Br_2_NO_2_ [M + H]^+^: 357.90783, found 357.90736.**4,4-Dibromo-2-(naphthalen-2-ylmethyl)isoquinoline-1,3(2H,4H)-dione (3aj).** White solid, 164 mg; mp: 112 °C; IR (KBr) cm^−1^ 3043, 1718, 1677, 1599, 1431, 1336, 1235, 1151, 975, 748, 597; ^1^H NMR (400 MHz, CDCl_3_) *δ* 8.15 (dt, *J* = 7.8, 1.7 Hz, 2H), 7.99–7.94 (m, 1H), 7.87–7.77 (m, 3H), 7.74 (td, *J* = 7.7, 1.4 Hz, 1H), 7.61 (dd, *J* = 8.5, 1.8 Hz, 1H), 7.53 (td, *J* = 7.7, 1.1 Hz, 1H), 7.50–7.40 (m, 2H), 5.43 (s, 2H). ^13^C NMR (100 MHz, CDCl_3_) *δ* 165.1, 161.6, 140.2, 134.6, 132.9, 132.8, 132.6, 130.4, 130.1, 128.3, 128.1, 127.9, 127.7, 127.3, 126.4, 125.9, 125.8, 120.1, 45.0; ESI-HRMS: *m/z* calcd for C_20_H_13_Br_2_NO_2_Na [M + Na]^+^: 479.92107, found 479.92071.**2-Benzyl-4,4-dibromo-6-methylisoquinoline-1,3(2H,4H)-dione (3ba).** White solid, 193 mg; mp: 105 °C; IR (KBr) cm^−1^ 2923, 1717, 1676, 1608, 1419, 1374, 1348, 1246, 971, 716, 699; ^1^H NMR (400 MHz, CDCl_3_) *δ* 8.01 (d, *J* = 8.1 Hz, 1H), 7.95–7.90 (m, 1H), 7.51–7.44 (m, 2H), 7.37–7.22 (m, 4H), 5.24 (s, 2H), 2.51 (s, 3H). ^13^C NMR (100 MHz, CDCl_3_) *δ* 165.1, 161.5, 145.9, 140.1, 135.5, 131.5, 130.4, 128.6, 128.3, 128.3, 127.6, 117.6, 44.7, 21.6; ESI-HRMS: *m/z* calcd for C_17_H_14_Br_2_NO_2_ [M + H]^+^: 421.93913, found 421.93872.**4,4-Dibromo-2-(3-bromobenzyl)-6-methylisoquinoline-1,3(2H,4H)-dione (3bb).** White solid, 198 mg; mp: 115 °C; IR (KBr) cm^−1^ 2917, 1724, 1675, 1608, 1433, 1331, 1225, 1152, 956, 685; ^1^H NMR (400 MHz, CDCl_3_) *δ* 8.01 (d, *J* = 8.1 Hz, 1H), 7.94 (s, 1H), 7.61 (d, *J* = 1.9 Hz, 1H), 7.40 (dd, *J* = 7.9, 1.8 Hz, 2H), 7.34 (dd, *J* = 8.1, 1.6 Hz, 1H), 7.19 (t, *J* = 7.9 Hz, 1H), 5.19 (s, 2H), 2.52 (s, 3H). ^13^C NMR (100 MHz, CDCl_3_) *δ* 165.1, 161.5, 146.1, 140.1, 137.6, 131.6, 130.8, 130.4, 129.8, 128.4, 127.2, 122.3, 117.4, 44.1, 21.6; ESI-HRMS: *m/z* calcd for C_17_H_12_Br_3_NO_2_ Na [M + Na]^+^: 521.83159, found 521.83128.**4,4-Dibromo-6-methyl-2-(3-methylbenzyl)isoquinoline-1,3(2H,4H)-dione (3bc).** White solid, 178 mg; mp: 92 °C; IR (KBr) cm^−1^ 2922, 1721, 1676, 1609, 1422, 1369, 1341, 1248, 1158, 971, 730, 696; ^1^H NMR (400 MHz, CDCl_3_) *δ* 8.01 (d, *J* = 8.1 Hz, 1H), 7.93 (s, 1H), 7.38–7.16 (m, 4H), 7.08 (d, *J* = 7.4 Hz, 1H), 5.21 (s, 2H), 2.51 (s, 3H), 2.32 (s, 3H). ^13^C NMR (100 MHz, CDCl_3_) *δ* 165.1, 161.5, 145.9, 140.2, 137.9, 135.4, 131.5, 130.4, 129.2, 128.4, 128.3, 128.2, 125.6, 117.7, 44.7, 21.6, 21.1; ESI-HRMS: *m/z* calcd for C_18_H_15_Br_2_NO_2_Na [M + Na]^+^:2 457.93672, found 457.93633.**2-Benzyl-4,4,6-tribromoisoquinoline-1,3(2H,4H)-dione (3bd).** White solid, 218 mg; mp: 112 °C; IR (KBr) cm^−1^ 3063, 1721, 1679, 1585, 1406, 1344, 1239, 1159, 971, 727, 693; ^1^H NMR (400 MHz, CDCl_3_) *δ* 8.28 (d, *J* = 1.8 Hz, 1H), 7.99 (d, *J* = 8.4 Hz, 1H), 7.66 (dd, *J* = 8.4, 1.8 Hz, 1H), 7.50–7.42 (m, 2H), 7.39–7.23 (m, 3H), 5.24 (s, 2H). ^13^C NMR (100 MHz, CDCl_3_) *δ* 164.4, 160.9, 141.6, 135.2, 133.9, 133.0, 129.8, 129.7, 128.7, 128.3, 127.7, 118.9, 45.0; ESI-HRMS: *m/z* calcd for C_16_H_11_Br_3_NO_2_ [M + H]^+^: 485.83399, found 485.83301.**4,4,6-Tribromo-2-(3-bromobenzyl)isoquinoline-1,3(2H,4H)-dione (3be).** Colorless oil, 219 mg; mp: 81 °C; IR (KBr) cm^−1^ 2921, 1723, 1683, 1586, 1472, 1403, 1336, 1236, 1158, 969, 756, 693; ^1^H NMR (400 MHz, CDCl_3_) *δ* 8.29 (d, *J* = 1.8 Hz, 1H), 7.99 (s, 1H), 7.68 (dd, *J* = 8.4, 1.8 Hz, 1H), 7.61 (q, *J* = 3.1, 2.5 Hz, 1H), 7.41 (td, *J* = 7.8, 1.7 Hz, 2H), 7.20 (t, *J* = 7.9 Hz, 1H), 5.19 (s, 2H). ^13^C NMR (100 MHz, CDCl_3_) *δ* 164.5, 160.9, 141.6, 137.3, 134.0, 133.1, 131.7, 131.0, 129.9, 129.9, 127.4, 122.4, 118.8, 44.3; ESI-HRMS: *m/z* calcd for C_16_H_9_Br_4_O_2_Na [M + Na]^+^: 585.72654, found 585.76965.**4,4,6-Tribromo-2-(3-methylbenzyl)isoquinoline-1,3(2H,4H)-dione (3bf).** Colorless oil, 200 mg; mp: 108 °C; IR (KBr) cm^−1^ 2922, 1720, 1682, 1586, 1405, 1339, 1234, 1161, 970, 759, 689; ^1^H NMR (400 MHz, CDCl_3_) *δ* 8.28 (d, *J* = 1.9 Hz, 1H), 7.99 (d, *J* = 8.4 Hz, 1H), 7.66 (dd, *J* = 8.4, 1.8 Hz, 1H), 7.32–7.11 (m, 4H), 7.09 (d, *J* = 7.6 Hz, 1H), 5.20 (s, 2H), 2.32 (s, 3H). ^13^C NMR (100 MHz, CDCl_3_) *δ* 164.5, 160.9, 141.6, 138.0, 135.1, 133.9, 133.0, 129.8, 129.7, 129.3, 128.7, 128.5, 128.2, 125.6, 119.0, 45.0, 21.1; ESI-HRMS: *m/z* calcd for C_17_H_12_Br_3_O_2_Na [M + Na]^+^: 521.83159, found 521.83128.**2-Benzyl-4,4,7-tribromoisoquinoline-1,3(2H,4H)-dione (3bg).** White solid, 223 mg; mp: 163 °C; IR (KBr) cm^−1^ 2922, 1702, 1670, 1588, 1432, 1348, 1259, 1148, 961, 696; ^1^H NMR (400 MHz, CDCl_3_) *δ* 8.27 (d, *J* = 2.1 Hz, 1H), 8.01 (d, *J* = 8.5 Hz, 1H), 7.85 (dd, *J* = 8.5, 2.1 Hz, 1H), 7.50–7.42 (m, 2H), 7.39–7.23 (m, 3H), 5.24 (s, 2H); ^13^C NMR (100 MHz, CDCl_3_) *δ* 164.6, 160.5, 139.0, 137.6, 135.1, 131.8, 131.0, 128.7, 128.4, 127.8, 124.9, 121.5, 45.1; ESI-HRMS: *m/z* calcd for C_16_H_10_Br_3_NO_2_Na [M + Na] ^+^: 507.81594, found 507.27163.**4,4,7-Tribromo-2-(3-methylbenzyl)isoquinoline-1,3(2H,4H)-dione (3bh).** Colorless oil, 199 mg; mp: 127 °C; IR (KBr) cm^−1^ 2925, 1683, 1599, 1430, 1320, 1297, 1245, 1020, 687; ^1^H NMR (400 MHz, CDCl_3_) *δ* 8.27 (d, *J* = 2.1 Hz, 1H), 8.01 (d, *J* = 8.5 Hz, 1H), 7.84 (dd, *J* = 8.5, 2.1 Hz, 1H), 7.26 (d, *J* = 9.1 Hz, 2H), 7.21 (t, *J* = 7.4 Hz, 1H), 7.09 (d, *J* = 7.4 Hz, 1H), 5.21 (s, 2H), 2.33 (s, 3H). ^13^C NMR (100 MHz, CDCl_3_) *δ* 164.5, 160.4, 139.0, 138.0, 137.6, 135.0, 131.8, 131.0, 129.3, 128.7, 128.5, 128.2, 125.6, 124.9, 121.5, 49.5, 45.0, 21.1; ESI-HRMS: *m/z* calcd for C_17_H_12_Br_3_O_2_Na [M + Na]^+^: 521.83159, found 521.83128.**2-Benzyl-4,4-dibromo-6-chloroisoquinoline-1,3(2H,4H)-dione (3bi).** Yellow solid, 88 mg; mp: 88 °C; IR (KBr) cm^−1^ 3087, 1730, 1688, 1587, 1426, 1374, 1349, 1264, 1230, 906, 733, 706; ^1^H NMR (400 MHz, CDCl_3_) *δ* 8.34 (dd, *J* = 7.6, 1.6 Hz, 1H), 7.85–7.72 (m, 2H), 7.52–7.46 (m, 2H), 7.35–7.23 (m, 4H), 5.21 (s, 2H). ^13^C NMR (100 MHz, CDCl_3_) *δ* 172.4, 161.0, 156.3, 137.2, 136.1, 135.3, 135.1, 132.0, 129.2, 128.9, 128.4, 127.9, 127.22, 44.3; ESI-HRMS: *m/z* calcd for C_16_H_10_ClNO_3_Na [M + Na]^+^: 322.02469, found 322.02361.**4,4-Dibromo-6-chloro-2-(3-methylbenzyl)isoquinoline-1,3(2H,4H)-dione (3bj).** White solid, 72 mg; mp: 108 °C; IR (KBr) cm^−1^ 2921, 1729, 1682, 1589, 1416, 1326, 1266, 1225, 1156, 1077, 976, 906, 764, 707; ^1^H NMR (400 MHz, CDCl_3_) *δ* 7.95 (dd, *J* = 7.7, 1.0 Hz, 1H), 7.77 (dd, *J* = 8.5, 7.8 Hz, 1H), 7.52–7.47 (m, 2H), 7.34–7.26 (m, 4H), 5.19 (s, 2H), 4.02 (s, 3H). ^13^C NMR (100 MHz, CDCl_3_) *δ* 172.2, 161.9, 160.8, 157.0, 136.7, 135.6, 131.5, 129.1, 128.3, 127.7, 121.9, 118.9, 117.3, 56.5, 44.1; ESI-HRMS: *m/z* calcd for C_17_H_14_NO_4_ [M + H]^+^: 296.09228, found 296.09301.**2-Benzyl-5-bromoisoquinoline-1,3,4(2H)-trione (3ca).** Colorless oil, 139 mg; mp: 158 °C; IR (KBr) cm^−1^ 2923, 1705, 1669, 157, 1427, 1361, 1237, 1163, 1025, 950, 733, 698; ^1^H NMR (400 MHz, CDCl_3_) *δ* 8.07 (dd, *J* = 8.9, 1.8 Hz, 1H), 7.57 (d, *J* = 2.4 Hz, 1H), 7.51–7.44 (m, 2H), 7.39–7.29 (m, 2H), 7.33–7.22 (m, 2H), 7.03 (dt, *J* = 8.8, 2.2 Hz, 1H), 5.23 (s, 2H), 3.96 (d, *J* = 1.8 Hz, 3H), 1.37–1.20 (m, 2H). ^13^C NMR (100 MHz, CDCl_3_) *δ* 165.1, 164.3, 161.2, 142.3, 135.6, 130.6, 128.6, 128.3, 127.5, 117.2, 114.3, 112.9, 55.7, 44.7, 29.4; ESI-HRMS: *m/z* calcd for C_17_H_13_NO_4_Na [M + Na]^+^: 318.07423, found 318.07315.**2-Benzyl-5-chloroisoquinoline-1,3,4(2H)-trione (3cb).** White solid, 81 mg; mp: 193 °C; IR (KBr) cm^−1^ 2922, 1700, 1674, 1574, 1433, 1362, 1332, 1235, 1153, 102, 753, 703; ^1^H NMR (400 MHz, CDCl_3_) *δ* 8.15 (dt, *J* = 7.8, 1.7 Hz, 2H), 7.99–7.94 (m, 1H), 7.87–7.77 (m, 3H), 7.74 (td, *J* = 7.7, 1.4 Hz, 1H), 7.61 (dd, *J* = 8.5, 1.8 Hz, 1H), 7.53 (td, *J* = 7.7, 1.1 Hz, 1H), 7.50–7.40 (m, 2H), 5.43 (s, 2H), 0.09 (s, 1H). ^13^C NMR (100 MHz, CDCl_3_) *δ* 165.1, 161.6, 140.2, 134.6, 132.9, 132.8, 132.6, 130.4, 130.1, 128.3, 128.1, 127.9, 127.7, 127.3, 126.4, 125.9, 125.8, 120.1, 45.0; ESI-HRMS: *m/z* calcd for C_16_H_10_ClNO_3_Na [M + Na]^+^: 322.02469, found 322.02414.**2-Benzyl-5-methoxyisoquinoline-1,3,4(2H)-trione (3cc).** Yellow solid, 100 mg; mp: 183 °C; IR (KBr) cm^−1^ 2924, 1723, 1680, 1587, 1453, 1379, 1358, 1283, 1225, 1070, 1014, 751, 699; ^1^H NMR (400 MHz, CDCl_3_) *δ* 8.39 (dd, *J* = 7.8, 1.1 Hz, 1H), 8.04 (dd, *J* = 8.1, 1.1 Hz, 1H), 7.66 (t, *J* = 7.9 Hz, 1H), 7.52–7.47 (m, 2H), 7.36–7.27 (m, 3H), 5.22 (s, 2H). ^13^C NMR (100 MHz, CDCl_3_) *δ* 165.6, 163.2, 160.8, 141.9, 140.7, 135.3, 133.94, 132.7, 130.8, 129.3, 128.3, 128.2, 127.9, 127.5, 125.3, 120.0, 44.7; ESI-HRMS: *m/z* calcd for C_16_H_10_BrNO_3_Na [M + Na]^+^: 365.97418, found 365.97343.**2-Benzyl-6-methoxyisoquinoline-1,3,4(2H)-trione (3cd).** Colorless oil, 123 mg; mp: 135 °C; IR (KBr) cm^−1^ 2924, 1695, 1674, 1593, 1466, 1286, 1008, 872, 757, 622; ^1^H NMR (400 MHz, CDCl_3_) *δ* 8.07 (d, *J* = 8.8 Hz, 1H), 7.57 (d, *J* = 2.5 Hz, 1H), 7.26 (d, *J* = 9.0 Hz, 2H), 7.20 (t, *J* = 7.5 Hz, 1H), 7.11–7.00 (m, 2H), 5.20 (s, 2H), 3.96 (s, 3H), 2.32 (s, 3H). ^13^C NMR (100 MHz, CDCl_3_) *δ* 165.1, 164.3, 161.2, 143.2, 137.9, 135.5, 135.7, 130.7, 129.2, 128.3, 128.2, 125.5, 117.2, 114.3, 113.0, 55.7, 44.7, 26.7; ESI-HRMS: *m/z* calcd for C_17_H_13_NO_4_Na [M + Na]^+^: 332.08988, found 332.09010.**2-(3-Bromobenzyl)-6-methoxyisoquinoline-1,3,4(2H)-trione (3ce).** White solid, 121 mg; mp: 102 °C; IR (KBr) cm^−1^ 2967, 1723, 1677, 1604, 1499, 1432, 1336, 1372, 1288, 1245, 1152, 1026, 960, 740, 691; ^1^H NMR (400 MHz, CDCl_3_) *δ* 8.07 (dd, *J* = 8.8, 1.8 Hz, 1H), 7.57 (d, *J* = 2.4 Hz, 1H), 7.54–7.46 (m, 2H), 7.36–7.25 (m, 3H), 7.03 (dd, *J* = 8.8, 2.4 Hz, 1H), 5.23 (s, 2H), 3.96 (d, *J* = 1.7 Hz, 3H). ^13^C NMR (100 MHz, CDCl_3_) *δ* 165.1, 164.3, 161.2, 142.3, 135.6, 130.6, 128.6, 128.3, 127.5, 117.2, 114.3, 112.9, 55.7, 44.7. ESI-HRMS: *m/z* calcd for C_17_H_13_NO_4_Na [M + Na]^+^: 318.07423, found 318.07493.**6-Methoxy-2-(3-methylbenzyl)isoquinoline-1,3,4(2H)-trione (3cf).** Colorless oil, 148 mg; mp: 112 °C; IR (KBr) cm^−1^ 2923, 1653, 1600, 1457, 1403, 1254, 1206, 1130, 1093, 1024, 944, 766; ^1^H NMR (400 MHz, CDCl_3_) *δ* 8.02 (d, *J* = 8.9 Hz, 1H), 7.56 (dd, *J* = 11.6, 2.1 Hz, 2H), 7.36 (t, *J* = 6.5 Hz, 2H), 7.15 (t, J = 7.8 Hz, 1H), 7.01 (dd, *J* = 8.9, 2.5 Hz, 1H), 5.15 (s, 2H), 3.92 (s, 3H). ^13^C NMR (100 MHz, CDCl_3_) *δ* 164.9, 164.3, 160.9, 142.1, 137.7, 131.3, 130.6, 130.5, 129.7, 127.1, 122.1, 117.1, 114.3, 112.6, 55.6, 50.7, 43.84; ESI-HRMS: *m/z* calcd for C_17_H_12_BrNO_4_Na [M + Na]^+^: 395.98474, found 395.98453.**2-Benzyl-7-methoxyisoquinoline-1,3,4(2H)-trione (3cg).** Colorless oil, 113 mg; mp: 109 °C; IR (KBr) cm^−1^ 2925, 1708, 1675, 1603, 1497, 1429, 1376, 1341, 1280, 1244, 1115, 1026, 961, 763, 698; ^1^H NMR (400 MHz, CDCl_3_) *δ* 7.99 (dd, *J* = 8.0, 1.2 Hz, 1H), 7.83 (dd, *J* = 7.8, 1.7 Hz, 1H), 7.52–7.45 (m, 2H), 7.45–7.30 (m, 4H), 7.14 (td, *J* = 7.7, 1.8 Hz, 1H), 5.39 (s, 2H). ^13^C NMR (100 MHz, CDCl_3_) *δ* 165.9, 141.0, 135.1, 134.5, 132.4, 130.7, 128.3, 128.2, 128.1, 127.6, 93.9, 67.0; ESI-HRMS: *m/z* calcd for C_16_H_11_Br_2_NONa [M + Na]^+^: 391.92855, found 391.92857.

### 4.3. X-ray Crystallographic Data of ***3aa***

The crystal of **3aa** for XRD analysis was prepared (see the Appendix A for details). CCDC 2217215 containing the supplementary crystallographic data can be obtained free of charge from the Cambridge Crystallographic Data Centre via www.ccdc.cam.ac.uk/data_request/cif.

## 5. Conclusions

In summary, we have demonstrated the first example of a direct oxidative coupling reaction of isoquinolines using inexpensive DMP. The methodology furnishes a diverse collection of synthetically valuable isoquinolinone under metal-free, easy-to-operate, and mild conditions. This method features a broad substrate scope with excellent functional group tolerance, affording structurally diverse isoquinoline-1,3-dione in high yields. Meanwhile, allyl bromide and 2-(bromomethyl) naphthalene were also suitable bromides to obtain isoquinoline-1,3-dione. Moreover, efficient access to isoquinoline-1,3,4-trione is also accomplished under the same conditions, reflecting this method’s synthetic utility. Further investigation into the biological activity of the above isoquinolinone and the synthesis of 4-bromoisoquinoline-1,3-dione is underway.

## Figures and Tables

**Figure 1 molecules-28-00923-f001:**
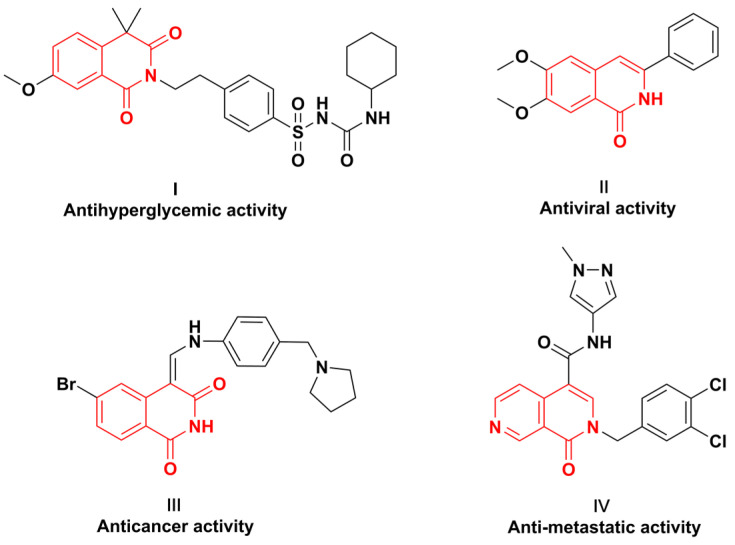
Selected biologically active compounds containing the isoquinolinone structural motif.

**Figure 2 molecules-28-00923-f002:**
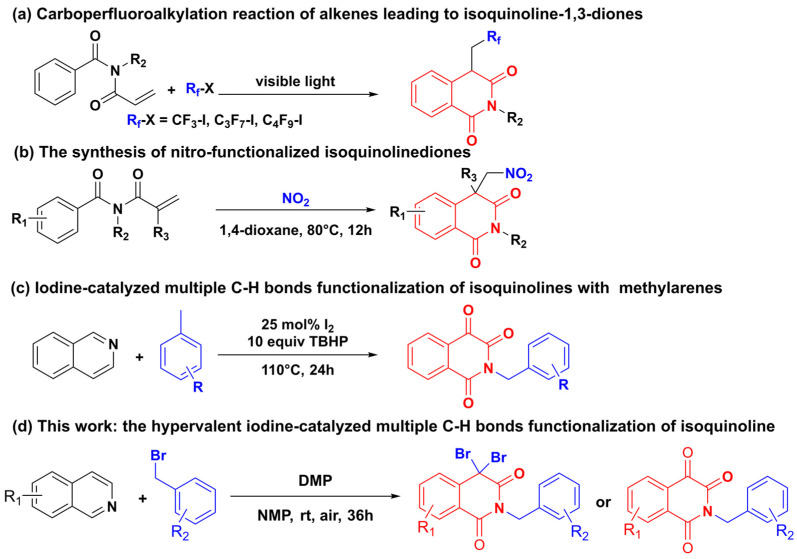
Various synthetic routes for isoquinolinone derivatives.

**Figure 3 molecules-28-00923-f003:**
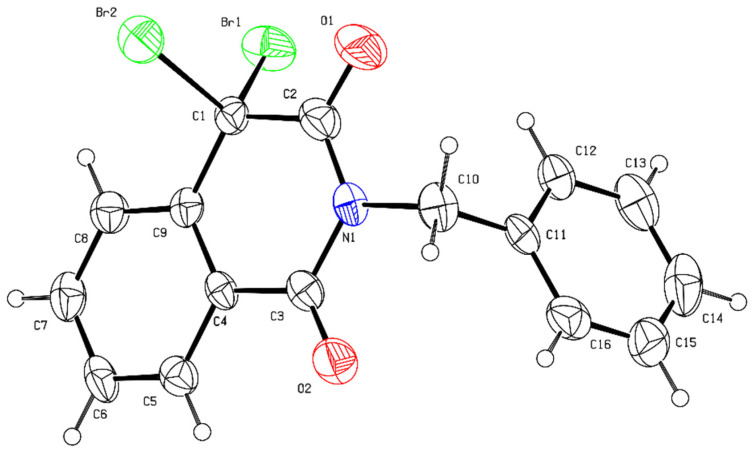
X-ray crystal structure of **3aa**.

**Figure 4 molecules-28-00923-f004:**
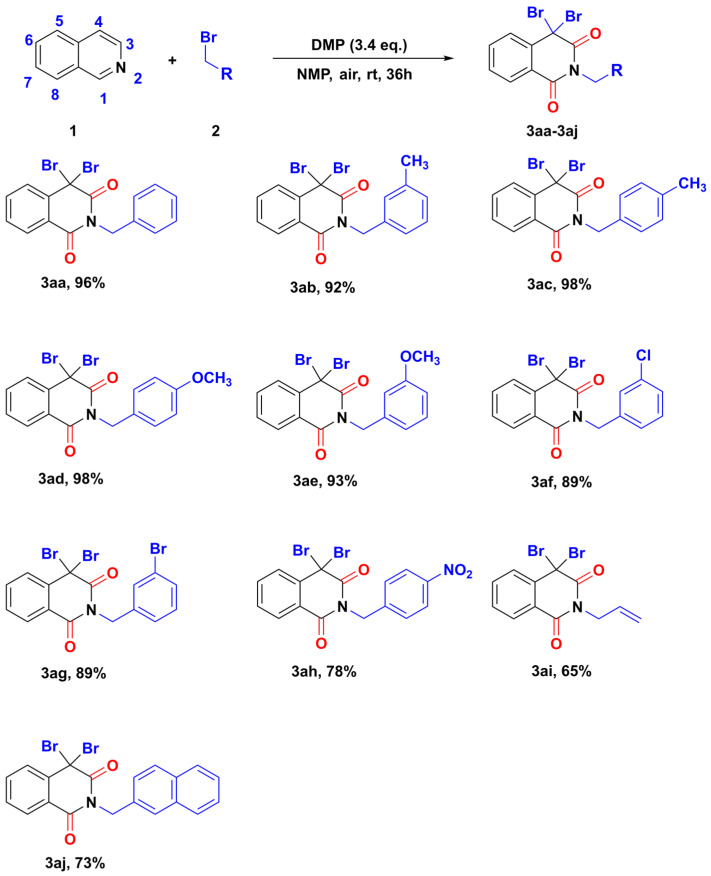
Investigating benzyl bromide substrates in isoquinoline oxidation coupling reaction.

**Figure 5 molecules-28-00923-f005:**
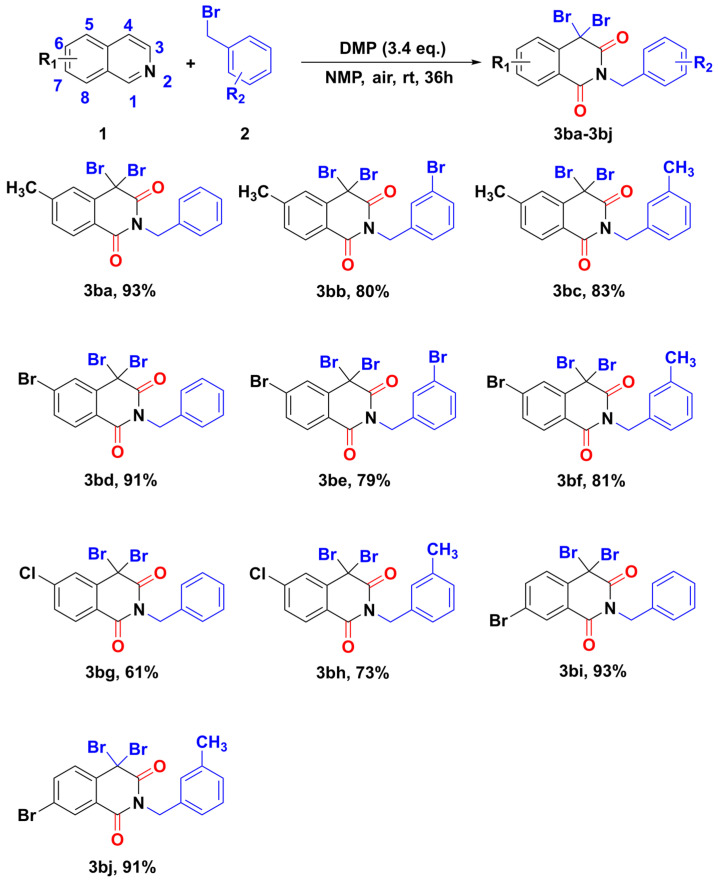
Substrate scope for isoquinoline counterpart.

**Figure 6 molecules-28-00923-f006:**
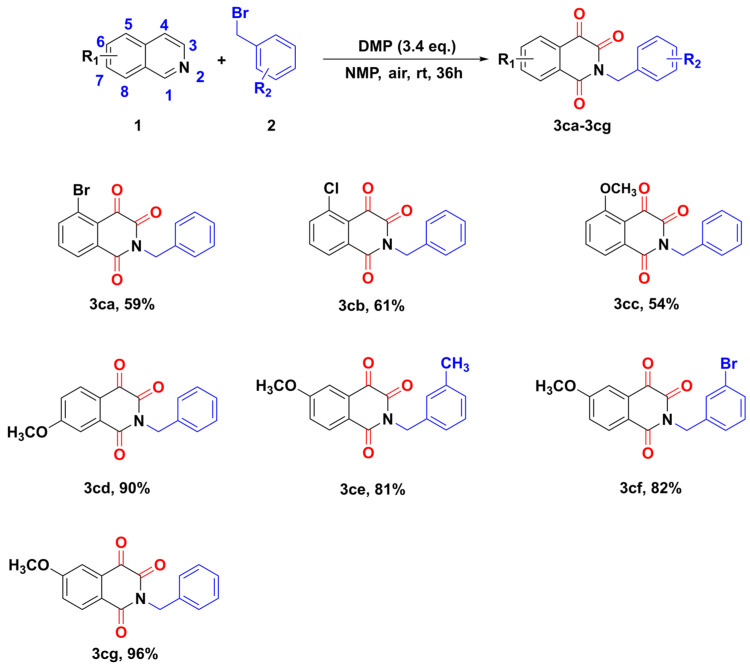
Scope of the synthesis of isoquinoline-1,3,4-triones.

**Figure 7 molecules-28-00923-f007:**
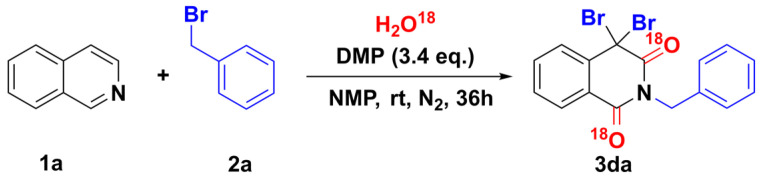
Mechanistic studies.

**Figure 8 molecules-28-00923-f008:**
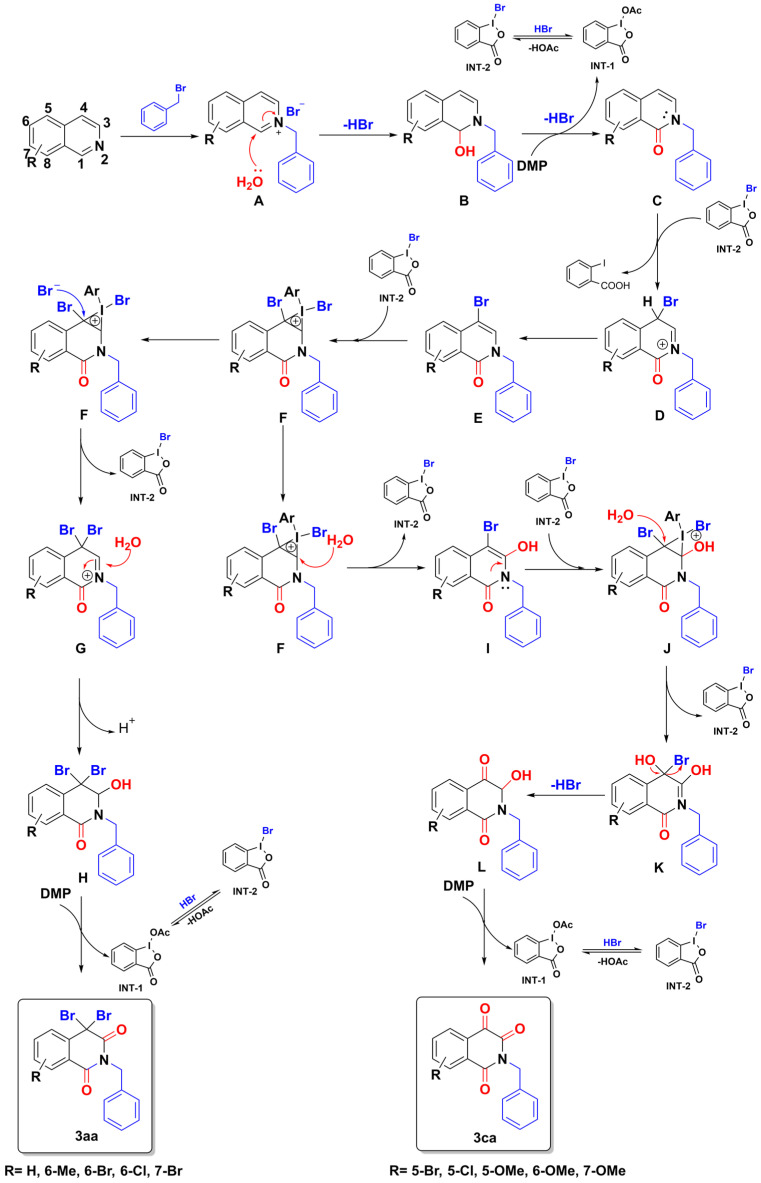
A plausible mechanism for the oxidative coupling reaction of isoquinoline mediated by DMP.

**Table 1 molecules-28-00923-t001:** Optimization of the reaction conditions ^a,b^.

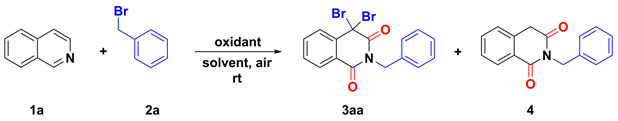
**Entry**	**Oxidant (Eq.)**	**Conditions**	**Yield (%) of 3aa/4**
1	DMP (2.0)	NMP, rt, air, 24 h	62/trace
2	PIDA (2.0)	NMP, rt, air, 24 h	0/0
3	PIFA (2.0)	NMP, rt, air, 24 h	0/0
4	Koser’s reagent (2.0)	NMP, rt, air, 24 h	0/0
5	PhIO (2.0)	NMP, rt, air, 24 h	0/0
6	IBX (2.0)	NMP, rt, air, 24 h	47/0
7	DMP (2.0)	DMF, rt, air, 24 h	38/0
8	DMP (2.0)	MeCN, rt, air, 24 h	55/0
9	DMP (2.0)	DMSO, rt, air, 24 h	0/0
10	PIDA (2.0)	DMF, rt, air, 24 h	0/37
11	DMP (2.0)	NMP, −10 °C, air, 24 h	Trace/0
12	DMP (2.0)	NMP, 0 °C, air, 72 h	39/0
13	DMP (2.0)	NMP, 50 °C, air, 24 h	18/0
14	DMP (1.0)	NMP, rt, air, 24 h	21/0
15	DMP (3.4)	NMP, rt, air, 36 h	96/0
16	DMP (3.4)	NMP, rt, N_2_, 36 h	0/0
17	DMP (3.4)	NMP, rt, air, H_2_O (10 eq.), 36 h	54/0

^a^ Reaction conditions: **1a** (1.0 eq., 0.1 mmol) and **2a** (2.5 eq., 0.25 mmol) in solvent (6.0 mL) at room temperature. ^b^ Yield was determined by NMR.

## Data Availability

The details of the data supporting the report results in this research, were included in the paper and Appendix A.

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
