# Peer review of "Dess–Martin Periodinane-Mediated Oxidative Coupling Reaction of Isoquinoline with Benzyl Bromide"

_molecules, 2023, doi:10.3390/molecules28030923_

Round 1

Reviewer 1 Report

This manuscript entitled “Dess–Martin Periodinane Mediated Oxidative Coupling Reaction of Isoquinoline with Benzyl Bromide” (2143853) by Wei Jiao et. al.  reports on the synthesis of N-substituted isoquinolinedione derivatives mediated by DMP.

This study seems to be progressive results based on the perspectives of development of new methodology, demonstrating first examples of direct coupling reactions of isoquinolines using DMP. Most of the reactions were well conducted and the diversity of this methodology should be considered and. Their mechanism studies were also meaningfully conducted.

However, some of the synthetic schemes and figures need to be revised. For an example, the structures in (a), Fig 1 need to be vertically flip so that the readers easily catch up as they are. Publication of this manuscript is recommended after minor revision. There are some concerns about the present manuscript which need to be addressed by the authors before publication: 

- Need to clarify the compounds are new or known. If known, the references should be put on the compounds including exp. section. If new, they have to add some more spectroscopic data such as HPLC, low-Mass, mp/bp, or IR to make full characterization. In general, six types of data are recommended for new compounds.  

- In Fig 1, the structure I should contain N in isoquinoline-dione !

- In Fig 2, the structures in (a) should be revised as discussed above. 

     In general, other drawings also need to be refined.

- In line 85, a space should be put as such: Table1, entry13 à Table 1, entry 13.

   There are so many mistakes like these. Please thoroughly check these.

- In line 92, the authors need to briefly mention about the by-products, if possible.

- The naming of the compounds has to have consistency. For examples, in line 116, 144 163, isoquinolinenone had better be changed into isoquinolinone. Please check the names in the whole manuscript.

Reviewer 2 Report

The manuscript entitled "Dess–Martin Periodinane Mediated Oxidative Coupling Reaction of Isoquinoline with Benzyl Bromide” is an interesting work by Chunmei Yang et al. describing the synthesis of twenty new isoquinoline-1,3-dione and seven isoquinoline-1,3,4-trione derivatives. The structures of new compounds have been confirmed by means of 1H NMR, 13C NMR, ESI-HRMS and also single-crystal X-ray diffraction (for compound 3aa). The manuscript is well written and pleasant to read. However, the following points need to be improved:

1. In Figure 1 structural formula of the compound I is incorrect, the nitrogen atom in the isoquinoline ring is missing.

2. Table 1 should be moved to line 68 because the description in lines 64-68 is unclear without this table.

3. In lines 69-70 there are abbreviations which need to be explain (PIDA, PIFA and PhIO). The abbreviations should be explained on first use or make a list of abbreviations in the beginning of the article.

4. All figures should be placed closer to the text that describes them.

5. In Figure 4:

·         there is R2 as a substituent, but there is no R1,

·         in the reaction scheme, the products have the symbol 3aa-3ag, but below there are also products 3ah, 3ai and 3aj, moreover compounds 3ai and 3aj does not fit to this scheme, please correct it.

6. In Figure 6, the symbol R1 under the compound formula is explained, but the symbol R2 is not explained. Please explain R2 as well.

7. In Figures 4 and 5, also under the formula, the symbols R1 and R2 should be explained.

8. There should be a larger gap between the text and the figures.

9. In chemical names of compounds there should be no gap before the name "isoquinoline" (in lines 204, 210, 216, 223, 229, 235, 241, 252, 270, 281, 287, 298, 309 and 345).

10. The Supporting Information should have numbered pages and a table of contents to make it easier to find a given compound.

11. In the Supporting Information there are spectra of three compounds (the last three) that are not in the text, please explain it.

12. In the Supporting Information there are spectra of compound which does not exist in the text (26th compound, isoquinoline-1,3,4-trione derivative with Cl in the C-6 position) and threre is a lack of spectra of two compounds (3bg i 3bh).

Reviewer 3 Report

Dear Authors,

Please address the following comments:

 1.      Page 2 line 42. Authors stated that “the direct syn-41 thesis of isoquinoline-1,3-dione from isoquinoline as the substrate has not been reported”. See the paper https://doi.org/10.1002/ejoc.202200645, reported the synthesis of Isoquinilone 1,3 dione from isoquinoline. Please correct the sentence and cite the manuscript.

 2.      Synthesis of dibromo derivative and trione derivative using the same reaction conditions? Please explain the difference between synthesis of dibromo derivative and trione derivative.

 Supplementary Materials:

     1.      Page 8, compound 3af: Compound needs to be repurified and revise the NMR spectra

2.      Page 10, compound 3ai: Compound needs to be repurified and revise the NMR spectra

3.      Page 13, compound 3bb: Compound needs to be repurified and revise the NMR spectra

4.      Page 16, compound 3be: Compound needs to be repurified and revise the NMR spectra

5.      Page 17, compound 3bf: Compound needs to be repurified and revise the NMR spectra

6.      Page 19, compound 3bi: Compound needs to be repurified and revise the NMR spectra

7.      Page 21, compound 3cc: Compound needs to be repurified and revise the NMR spectra

8.      Page 23, compound 3cg: Compound needs to be repurified and revise the NMR spectra

9.      Page 30: Compound needs to be repurified and revise the NMR spectra.

Round 2

Reviewer 2 Report

I thank the Authors for introducing the corrections I suggested. In my opinion the article now is more understandable and acceptable. However, there are some minor errors like:

·        - in Figure 6 in the reaction scheme, the reaction product has two carbonyl groups instead of three

·        -  in „Original image” in Figure 1 structural formula of the compound I is still incorrect, the nitrogen atom in the isoquinoline ring is missing

Author Response

Sorry to keep you waiting, we carefully examined the manuscript this time and made appropriate adjustments.
